# Machine Learning Algorithms Applied to Identify Microbial Species by Their Motility

**DOI:** 10.3390/life11010044

**Published:** 2021-01-12

**Authors:** Max Riekeles, Janosch Schirmack, Dirk Schulze-Makuch

**Affiliations:** 1Astrobiology Group, Center of Astronomy and Astrophysics, Technical University Berlin, 10623 Berlin, Germany; j.schirmack@tu-berlin.de (J.S.); dirksm@wsu.edu (D.S.-M.); 2Section Geomicrobiology, GFZ German Center for Geosciences, 14473 Potsdam, Germany; 3Department of Experimental Limnology, Leibniz-Institute of Freshwater Ecology and Inland Fisheries (IGB), 16775 Stechlin, Germany; 4School of the Environment, Washington State University, Pullman, WA 99164, USA

**Keywords:** machine learning, motility, biosignature, automation, species identification, life detection

## Abstract

(1) Background: Future missions to potentially habitable places in the Solar System require biochemistry-independent methods for detecting potential alien life forms. The technology was not advanced enough for onboard machine analysis of microscopic observations to be performed in past missions, but recent increases in computational power make the use of automated in-situ analyses feasible. (2) Methods: Here, we present a semi-automated experimental setup, capable of distinguishing the movement of abiotic particles due to Brownian motion from the motility behavior of the bacteria *Pseudoalteromonas haloplanktis, Planococcus halocryophilus, Bacillus subtilis, and Escherichia coli*. Supervised machine learning algorithms were also used to specifically identify these species based on their characteristic motility behavior. (3) Results: While we were able to distinguish microbial motility from the abiotic movements due to Brownian motion with an accuracy exceeding 99%, the accuracy of the automated identification rates for the selected species does not exceed 82%. (4) Conclusions: Motility is an excellent biosignature, which can be used as a tool for upcoming life-detection missions. This study serves as the basis for the further development of a microscopic life recognition system for upcoming missions to Mars or the ocean worlds of the outer Solar System.

## 1. Introduction

Every planetary mission so far has used optics for direct imaging at macroscopic scales. However, imaging was used mainly for geological purposes in these missions, and so microscopes on rovers or landers have not been employed to detect life [1]. One major problem with using microscopic in-situ observations to search for microbial organisms is that there are not many features that distinguish cells from abiotic sediment particles. The unambiguous distinction of primitive life forms such as bacteria and archaea from mineral particles even with an electron microscope is not always possible [2]. Thus, to detect life unambiguously, chemical composition or microbial motility has to be added as a criterion [1].

Microbial motility is the property of the movement of a cell using its own power. Motility is common to all three domains of life (bacteria, archaea, and eukaryotes) and is stimulated by environmental conditions [3]. This ability to move is crucial for many organisms because it enables them to avoid toxins and find new resources and nutrients. Many prokaryotes use flagella for propulsion. Flagella are filamentous appendages that rotate at up to 300 revolutions per second and propel the cell through liquid media at up to 60 cell lengths per second [4]. The amino acid sequence of flagellin—an essential protein of the flagella—is highly conserved in bacterial species, suggesting that flagellar motility evolved early and has deep phylogenetic roots. The fundamental differences of the flagellar protein structure in archaea and bacteria indicate that flagellar motility evolved separately in these two prokaryotic cell types, presumably after they diverged over 3.5 billion years ago [4]. The main characteristic of bacterial motility is speed and the motion pattern. A quantitative description of the distribution of rotational motions and the speed in spatiotemporal space describes the bacterial movement and allows the characterization of different motility behaviors. The values of these parameters are characteristic not only for each specific species but also for the environmental conditions. Different forms of motility patterns have been qualitatively described by Reference [5]. The strains we used in our experiments show dominantly run-and-tumble motion (*Escherichia coli* and *Bacillus subtilis*), or a run-and-reverse-flick pattern (*Pseudoalteromonas haloplanktis* and *Planococcus halocryophilus*), but an extensive quantitative evaluation is missing. Previous research indicates that the average translational velocity of all bacteria seems to increase as the environmental temperature increases [6].

The main unknown question, assuming the existence of extraterrestrial microbial life, is whether it would move using its own propulsion (like using flagella or a similar system) and, if so, how it might move. Because motility is such an advantageous characteristic and one which evolved very early in the evolution of life on Earth, we suggest that it is likely that extraterrestrial life has developed this property as well [4]. If so, then the distinction between Brownian motion and microbial motility should be straightforward and is expected to have high accuracy when applying machine learning algorithms.

It would be ideal to make the distinction with a minimum set of parameters. This is important because one of the challenges of planetary missions is the constrained link budgets. Only limited amounts of data can be transmitted from a lander—either via a relay or directly—to Earth. Therefore, it is crucial that most of the data can be analyzed autonomously. It is also advantageous to determine beforehand which data should be transmitted autonomously and which not. An optimal engineering solution is a trade-off between the computing power required for the autonomous processes and the link budget, which requires detailed knowledge of the mission concept. The software solution needs to be a trade-off between false positive and false negative rates (depending on the link budget, a bias toward false positive might be preferable, but in any case, the ground operators should re-evaluate the transferred data). This study aims to identify life based on motility by using supervised machine learning algorithms, showing that identification of microbial species is possible by analyzing their motility and collecting detailed quantitative data, which complement the qualitative descriptions performed in previous attempts. Here, we address the identification of motility with tools that can be extended to use in space missions.

## 2. Materials and Methods

### 2.1. Microorganisms Used

Aside from the well-known *Escherichia coli* bacterium, which has been commonly used in many previous experiments, we have also tested our detection technique using *Bacillus subtilis*, *Planococcus halocryophilus*, and *Pseudoalteromonas haloplanktis*. *B. subtilis* is a common and resistant soil bacterium. *P. halocryophilus* and *P. haloplanktis* are adapted to salty and cold environments, making them suitable analog organisms for life detection on Mars and the ocean worlds such as Europa and Enceladus.

*Escherichia coli* is the most widely studied prokaryotic organism. It is a rod-shaped, facultative anaerobe gram-negative bacterium, which lives in the guts of warm-blooded organisms. The organisms are about 2 µm long, 1 µm in diameter, and are motile due to the possession of flagella [4]. It is a peritrichous bacterium, meaning it has 5–10 flagella, which are randomly distributed across the cell surface [7]. We used *E. coli* K-12 wild-type (DSM 498) in our study, obtained from Dirk Wagner, Helmholtz Centre Potsdam GFZ German Research Centre for Geosciences.

*Bacillus subtilis* is a facultative anaerobic bacterium and one of the best-studied gram-positive organisms. *B. subtilis* occurs in the natural soil environment and is also found in human’s gastrointestinal tract [8]. It is rod-shaped and can survive in extreme conditions in terms of temperatures and desiccation because it is able to form endospores. It is typically 4 to 10 µm long and 0.25 to 1 µm in diameter [9]. We used the type strain *B. subtilis* Marburg (DSM 10), strain designation Marburg, obtained from Dirk Wagner, Helmholtz Centre Potsdam GFZ German Research Centre for Geosciences. This strain moves by using approximately 20 flagella per cell, which are non-randomly distributed over the cell surface [10].

*Pseudoalteromonas haloplanktis* is a curved, rod-shaped psychrophilic bacterium isolated from Antarctica. It is typically 1.2 to 2.3 µm long and 0.5 to 0.6 µm wide. This strictly aerobic extremophile is gram-negative, non-spore-forming, and motile, employing a single polar flagellum [11]. It has its growth optimum at 26 °C and can grow at temperatures as low as 0 °C [12]. Due to its ability to survive in cold environments, we used the type strain *Pseudoalteromonas haloplanktis* 545 (DSM 6060), obtained from DSMZ (Leibniz Institute DSMZ-German Collection of Microorganisms and Cell Cultures GmbH).

*Planococcus halocryophilus* is a gram-positive bacterium. It is aerobic and found in the arctic permafrost. The cells are non-spore-forming cocci, which occur individually or in pairs, and have a diameter of between 0.8 and 1.2 µm. The bacterium is motile and thrives in environments of high salinity and low temperature [13]. It grows at temperatures as low as −10 °C and is metabolically active at temperatures as low as −25 °C [14]. Cells remain motile under all growth conditions [14]. The strain was first isolated from a permafrost active layer-soil in Ellesmere Island, Eureka, Canada. We used the type strain *Planococcus halocryophilus* Or1 (DSM 24743), obtained from DSMZ (Leibniz Institute DSMZ-German Collection of Microorganisms and Cell Cultures GmbH).

### 2.2. Incubation of Microorganisms

Before conducting the microscopic measurements at different temperatures, the microbial strains were allowed to grow to the late exponential phase. *E. coli* and *B. subtilis* were both grown for 24 h at 30 °C in standard nutrient broth S1. *P. haloplanktis* was incubated for 48 h at 25 °C in bacto marine broth (DIFCO 2216), and *P. halocryophilus* was grown for 48 h at 25 °C in trypticase soy yeast extract medium.

### 2.3. Measurements

We used a *Primo Star* Full Köhler phase contrast microscope with a 40× objective for the observations. Connected with the microscope was an *AxioCam 105* color camera, which transmitted the pictures live to a computer. The microscope in combination with the digital camera leads to a pixel scale of 0.11 µm restricted by the Abbe diffraction limit (approximately 0.4 µm). We used a Neubauer counting chamber with a depth of 10 µm for the observations of the microbial sample and of a milky solution as a reference sample. The microscopic measurements were performed at temperatures of 25 °C, 30 °C, 35 °C, and 40 °C. To maintain a stable temperature, we used a thermal insulating chamber consisting of 4 cm polystyrene walls and a 5 mm insulating aluminum layer. We used a *DOSTMANN P700 Universal Precision Thermometer.* The probe of the thermometer was attached to the slide of the microscope. The chamber enclosed the microscope and the camera. We used a *Philips ThermoProtect 2100 W* hairdryer for controlling the temperature in the chamber. The setup of the temperature control chamber can be seen in Figure 1.

Due to time constraints, we tracked 50 individual movement pathways over ten seconds for each species, which was approximately 9.4% of the total individual *E. coli* cells observed, 10% of the total individual *P. halocryophilus* cells, 12% of the total individual *P. haloplanktis* cells, and approximately 40% of the total individual *B. subtilis* cells. Exposure times varied between species (and therefore optimal lighting for best image quality). The obtained videos had between 6.94 and 5.41 frames per second (in each case, the shutter speed was fast enough that the microbes could be clearly seen in the individual images, without smearing out).

Video files (.tiff) were imported for tracking into the software Fiji/ImageJ. For this, we used the manual tracking plugin, for which we calibrated the time interval and the x/y calibration according to the parameters of the microscopic recordings. With the software, we obtained the speed and the x/y-coordinates of the microbes for each video frame by clicking on their center with the mouse pointer.

### 2.4. Errors

We obtained images of 2560 × 1920 pixels with our system. Due to the digital zoom in the Fiji software, we could determine the location of the microbes with a ±1 pixel (=0.11 µm) accuracy, which is, depending on the frames rate of the videos, equivalent to a ±0.62 µm/s accuracy for *E. coli* and *B subtilis*, ±0.73 µm/s for *P. halocryophilus,* and ±0.76 µm/s for *P. haloplanktis*. Another source of error in tracking is that we only tracked the microbes in two dimensions, even though they were in a 10 micrometer (about ten microbial body lengths) deep medium. This limitation applies to all observed microbes. Another potential error was that for observations of this kind, externally induced shifts by convection must be prevented. Therefore, we observed a reference sample (a milk solution in the motility medium) on the stage with the same configuration of heating source, thermal chamber walls, and microscope, and adjusted until no convective motion in the medium occurred before the start of each run.

### 2.5. Classes and Features of Data

We worked with aggregates (defined as the averaged properties of the observed cells over the length of the recordings) of features over the recorded time to avoid biases caused by methodological errors, such as variations of exposure time from recording to recording. At the same time, it is important to provide the tracking and classification algorithms only with information that does not give them a priori information about the observed species, such as by an increase in the occurrence of a specific species in a certain coordinate range. Therefore, no coordinate data was passed on to the algorithms.

Two classes were used to classify the cells and the abiotic particles with the supervised learning algorithms: The first class consisted of the four observed bacterial species at 25 °C, and the second class consisted of the simulated mobility traces due to Brownian motion at the same temperature. Subsequently, the cells were classified into four classes representing the used species.

The microbes and the simulated particles were tracked using individual motility/mobility traces over ten seconds. Depending on the species and exposure time, between 54 and 69 consecutive images were taken for each microbial pathway. Two consecutive images were needed to determine the speed and three consecutive images for the angle change (e.g., a ten-second image sequence of 54 consecutive images resulted in 53 speed information and 52 angle information units for each microbial pathway).

To ensure that the different exposure times do not provide the classifiers with a priori information, we determined the average values of the microbial pathway features over ten seconds. We intended to select properties that classify the motion as simple as possible to minimize the computational requirements. A too complex analysis of the motion behavior (including complex consecutive motion properties) would risk overfitting certain features that might work well for our data set but fail for some other data set. A general problem with classifications is that it is not known in advance what the most suitable features are for classification. In our case, this means: what are the motility features leading to the best accuracy of classifying microbial species, and how to classify best biotic vs. abiotic particles? The features we used were: (1) mean speed of the particle, (2) standard deviation of mean particle speed, (3) relative amount of clockwise directional change, (4) relative amount of counterclockwise directional change, (5) relative amount of low (<20°) directional change (6) average directional changing angle, (7) standard deviation of directional changing angles, (8) relative amount of low speed (set arbitrarily at < 1.4 µm/s and used as a tunable parameter), (9) mean distance of particles after ten seconds.

Measured angles during movement are taken from three consecutive coordinates based on the motility behavior of the microbes: The angles here are the angles that form the two sides $\bar{12}$ and $\bar{23}$. 1 is the coordinate of the microbe in the first video frame, 2 in the following one, and 3 in the one thereafter (see Figure 2).

Besides the features (1), (2), (6), and (7) (mean speed and angles, respectively, and their standard deviations), which are the most apparent properties to quantify the microbial pathways, there are additional ways to quantify motility patterns. Bacteria moving near surfaces experience hydrodynamic forces that attract them towards the surface and cause them to move in circular trajectories [15]. Due to this phenomenon, we also quantified the direction of the feature with (3) and (4). In feature (5), we defined a directional change of <20° as a straight movement. The definition of low speed <1.4 µm/s is added as a feature (8). This is an empirical parameter determined after the observations were concluded, and it is suitable for classification because of its statistically significant variations between bacterial species. The speed of 1.4 µm/s equals approximately the mean displacement per second of a particle with a diameter of 0.5 µm at 25 °C. Feature (9) is a way to easily quantify the straightness of an object’s average movement over the observation period.

It is crucial to select features that contribute most to the predicted output. Irrelevant features in the data can decrease the accuracy of the classification. Feature selection means less redundant data and, therefore, less impact from noise, more accurate models, and faster training of algorithms. To remove irrelevant features, we implemented an exhaustive feature selector for sampling and evaluating all possible feature combinations to classify the species. Calculating all nine features from the observed coordinates of the respective frames required very little computer power.

We used 10-fold cross-validation for all algorithms, during which the dataset was shuffled randomly and split into ten groups. Each group was used once as a test group, with the remaining groups being used as training data. A model fitting the training set was evaluated in the test set. The result of the evaluation was kept. The model was then discarded, and the next group was used as a test group. Then, the quality of the algorithm was evaluated by the respective individual test results. The accuracy of the algorithm was then determined for all test groups. This increases the accuracy of estimating how the models are expected to perform with data not used during the training of the model. We used Python 3.7 in the scientific environment Spyder for running the algorithms. The classifiers were Logistic Regression Classifiers (LRC), Linear Discriminant Analysis (LDA), K-Nearest Neighbor Classifiers (KNN), Classification and Regression Trees (CART), Naïve Bayes (NB), and Support Vector Machines (SVM). For detailed information on these classifiers, see Appendix A and [16].

### 2.6. Simulation of Brownian Motion

Einstein (1905) developed a statistical mechanics theory for Brownian motion [17]. It states that the mean displacement of a Brownian particle is proportional to the square root of the elapsed time. That is where the mean displacement squared is $\bar{x^{2}}$, the diffusivity is D, and the elapsed time is t. The diffusivity is given by:(1)$$\bar{x^{2}}=2Dt$$
(2)$$D=\mu k_{B}T$$
with $k_{B}=1.3806\times 10^{-23}$ (Boltzmann’s constant), T being the temperature, and μ being the ratio of the particle’s drift speed to an applied force, which can be calculated by $\mu =\frac{1}{6\pi \eta r}$. η is the dynamic viscosity of the fluid, and r is the particle’s radius. This leads to:(3)$$D=\frac{k_{B}T}{6\pi \eta r}$$

Based on these equations, we simulated the movements of particles due to Brownian motion at the temperatures investigated with Matlab. We ran simulations with viscosities for water at 25 °C (η = 0.89 mPa*s) assuming spherical particles, with radii between 0.25 µm and 1 µm. The advantage of the simulations is that one can easily vary all relevant parameters. An additional benefit is that massive data sets can be quickly generated to train supervised learning procedures.

## 3. Results

### 3.1. Motility Parameters

In the simulations, the mean speed due to Brownian motion was 3.03 $\frac{µm}{s}$ at 25 °C with a standard deviation of $0.78\frac{µm}{s}$. The mean distance a particle traveled during ten seconds was 4 µm with a standard deviation of 2.5 µm, and the relative amount of low speed (<1.4 µm/s) was 18% with a standard deviation of 9%.

The observed mean speeds of the tracked bacteria varied with microbial species. *E. coli* was the fastest observed among all bacteria, with a mean speed of $7.59\pm 0.62\frac{µm}{s}$ at 25 °C. The mean speed of *P. haloplanktis* was measured to be $5.30\;\frac{µm}{s}\pm 0.76\frac{µm}{s}$ at 25 °C and *P. halocryophilus* had a mean speed of $5.21\pm 0.73\frac{µm}{s}$ (Figure 3a and Appendix A). *B. subtilis* were moving the fastest and had a mean speed of $6.89\;\frac{µm}{s}\pm 0.62\frac{µm}{s}$ at 25 °C. (Figure 3b). At 25 °C, the movements due to (the simulated) Brownian motion were clearly slower than the microbial movements.

We heated the liquid water medium and observed the bacterial species additionally at 30 °C, 35 °C, and 40 °C. The change of mean speed with temperature was variable for the different species tested. *E. coli* was the only species that had its highest speed at 40 °C. All other bacteria tested had no consistent behavior with regard to an increase in temperature. However, as expected, (the simulated) Brownian motion increased with temperature. For *P. halocryophilus* and *P. haloplanktis*, we observed a similar relationship of mean speed as function of temperature. However, we noticed that both microbial species had speeds similar to (the simulated) Brownian motion at increased temperatures, which was likely the case because these elevated temperatures were out of their viability range, and the cells were not anymore self-propelled but dead (see Figure 3a).

The speeds of the microbes indicated large standard deviations. The mean distance from the starting point after ten seconds and the relative amount of small speed also showed high standard deviations (Figure 4a,b). The distance from the starting point for each microbial pathway and the relative amount of small speed are shown in Figure 5 and Figure 6. Comparisons to normal distribution, histograms of the microbial speed distribution, and Q-Q plots are shown in the Appendix A.

The mean distance that the microbes traveled after ten seconds at 25 °C correlated well with the respective mean speeds of the microbes tested (e.g., the Pearson correlation coefficient was 0.83). *E. coli* achieved a mean distance of 41.70 µm and *B. subtilis* a mean distance of 32.67 µm. *P. halocryophilus* advanced after ten seconds 14.59 µm, and *P. haloplanktis* 30.22 µm. *P. haloplanktis* and *P. halocryophilus* had approximately the same mean speed, but *P. halocryophilus* covered less distance than *P. haloplanktis*, which likely means it made more turns and stops during its movement.

The mean relative amount of low speed (<1.4 µm/s) at 25 °C was highest for *P. haloplanktis* and *P. halocryophilus* with 8.0% and 8.1%, respectively. *B. subtilis* exhibited a mean relative amount of low speed of 5.8% and *E. coli* of 3.5% (Figure 6, mean values for the simulated pathways are also included in the figure).

Our observations of the features “standard deviation of particle speed”, “relative amount of low directional change (<20°)”, “average directional angle”, and “standard deviation of directional changing angle” did not show any substantial differences on the species level and thus were not substantial enough to be useful for the automated classification.

### 3.2. Results of Classification Algorithms

All classification algorithms were shown to have higher detection accuracy than 98% for distinguishing between observed bacteria and the simulated Brownian motion (Figure 7a). The best accuracy is that of the classification algorithm based on Support Vector Machines, Naïve Bayes, and K-Nearest Neighbor (with K = 1) (99.75% ± 0.25% classification accuracy). Six of the seven classifiers reveal the “mean distance after ten seconds” tracking parameter as the most successful classification scheme when only one parameter is used. Only the Linear Discriminant Analysis Classifier achieved a better classification accuracy with the feature mean speed.

The classification accuracies for the assignments of the four bacterial species are shown in Figure 7b, which highlight the usefulness of the algorithms with the optimum feature selection. The best classification accuracy has the K-Nearest Neighbor Classifier (KNN, with K = 1) with an accuracy of 82% ± 5.7%.

The optimal feature for the highest detection accuracy and the second-highest detection accuracy was the relative amount of low speed (Table 1). The KNN Classifier and the CART Classifier were not only with their optimal feature selection (relative amount of low speed) superior to the other algorithms but also with their respective second-, third-, and fourth best feature selection.

We note that especially for the algorithms with high classification accuracy, i.e., KNN and CART, only a few features were selected. The KNN Classifier (with K = 1, see Appendix A) used for its most successful classification results the features relative amount of low speed, relative amount of clockwise directional change, relative amount of counterclockwise directional change, and relative amount of low directional change. The CART Classifier, which delivered the second-best classification results, used the features relative amount of low speed, average directional angle, and relative amount of counterclockwise directional change (Table 1).

## 4. Discussion and Conclusions

As shown in Figure 3, Figure 4, Figure 5 andFigure 6, there is a substantial amount of noisiness in the data. Our objective was to discriminate life from non-life using multiple independent criteria. The discrimination is challenging because it is difficult to apply a weighting procedure and motility is not easily traceable to basic physical principles. However, our results show that it is possible to differentiate between the movement of microbial cells and abiotic particle movements (simulated Brownian motion) with very high confidence and accuracy of > 99%. This indicates that Machine Learning is very helpful for this kind of data processing. All tested algorithms successfully distinguished biotic motions from the simulated movements of abiotic particles of similar size. The mean square displacement of particles due to Brownian motion was proportional to the elapsed time, which made the mean distance of particles traveled after ten seconds the most important feature for this purpose. However, it was challenging to distinguish between different bacterial species due to variations in the investigated features, which was observed for all four tested species. The distance advanced by the microbial species after ten seconds correlated only to a limited degree to their respective mean speeds. A feature that seems to be superior for the classification of the bacterial species was the relative amount of low speed (which is a tunable parameter and best classification results were achieved for speeds of <1.4 µm/s). This feature was particularly suitable for the classification of bacterial species when using the KNN or the CART algorithm.

All of the used features for the classification (biotic vs. abiotic, and species identification) are simple statistical values, which can be determined relatively easily automatically without high computational power through two-dimensional observations of the motility pathways. Thus, a quantitative analysis of the movement of microbes may also be a way to screen samples for certain specific microbes if sufficient quantitative data on microbial motility are available.

3D-microscopy approaches used to track microbes like digital holographic microscopy or defocused image methods are a trade-off between performance and ease of use [18,19]. Microbial motility occurs in nature in three dimensions, and a 3D observation most likely will increase the identification accuracy. Thus, it will be instructive to conduct similar experiments, but in 3D, in the future. Nevertheless, the classification with our more straightforward 2D approach with much less data processing still provided acceptable accuracies and results, especially when distinguishing microbial motility from Brownian motion of particles similar in size to microbes. A trade-off between technical requirements must be considered when designing space missions to search for life in extraterrestrial bodies such as Mars or Europa. The maximal data volume capability of NASA’s Europa-Lander for its microscope is 302 Mbit/s. Ten-second recordings of our observations have a data size of approximately 3600 Mbit. This shows that the transmission of longer recordings, which will undoubtedly be necessary, is impractical, at least with our current laboratory hardware. The output of all the processed data, however, has a data size of less than 9 Mbit. Thus, if the data could be processed onboard [20] rather than being transferred to ground control and analyzed on Earth, then this is a feasible way to conduct the motility analyses.

Liquid water is a necessary condition for the existence of life as we know it [21]. Thus, our experiments to expose microbes to liquid water as a medium is appropriate. However, during a space mission to Mars or the subsurface oceans of the icy satellites, this may pose an additional challenge because of the possible instability of the liquid water sample during microscopic observations. To observe the motility of particles (and possible microbes) within the sample, it is crucial that the system does not exhibit any movements induced by external forces or internal forces, for example, vibrations of the system due to movements of actuators on the lander. Additionally, it is conceivable that magnetic forces could interact with the sample, provided it contains ferromagnetic elements. Possible external electromagnetic forces or forces induced by components in the lander could also lead to unwanted movements in the sample. Therefore, the microscopic system must have sufficient shielding. Alternatively, it may be possible to subtract the effect of external motions from the microbial motility behavior. This seems to be a feasible approach for background laminar flow after more insights about the interaction of bacteria and fluid flow are gained because laminar flow equations are well known and understood. This will be more challenging for other external forces such as background turbulent flow or magnetic fields that fluctuate with time.

Thermal stability is one of the most crucial aspects that need to be considered when conducting a microscopic detection of motility. Under specific scenarios, for example, when a frozen sample is obtained from an icy moon, it may be necessary to heat the sample. The heating process is dynamic, and the Brownian motion of any liquid will accelerate during heating. Uneven heating of the solvent (and possible evaporation) will cause a flow that can affect any microbes in the fluid and their movements. Microbes that could be detected under “stable” thermal conditions could be non-distinguishable from inanimate matter when there is a bulk flow forced by heat transfer (convection).

Viscosity, temperature, and particle size are also important. To make quantitative comparisons to Brownian motion, accurate information is needed about the sample’s viscosity, the temperature, and the particle sizes observed. The latter can be done using a filter system that only allows certain particle sizes to pass into the liquid sample. Viscosity and temperature must be determined in-situ.

We have observed that microbes behave differently at different temperatures. This is also true for other stimuli (chemical, light). This characteristic behavior for each species could be exploited for a microscopic life detection device when combined with machine learning techniques. The variability of the environmental conditions, not only the temperature but also inducing chemical microbial stimuli, is worthy of further investigation. We expect that we might be able to observe putative microbes moving toward or away from the provided stimulus. Morphological observations may be an additional criterion of whether life is indeed present. The result of this further research may eventually allow the design of a motility-based instrument suitable as a life-detection device. Such a device might not only be suitable for space missions but also for microbial species identification on Earth, for example, to distinguish pathogenic from non-pathogenic microorganisms in a water sample.

## Figures and Tables

**Figure 1 life-11-00044-f001:**
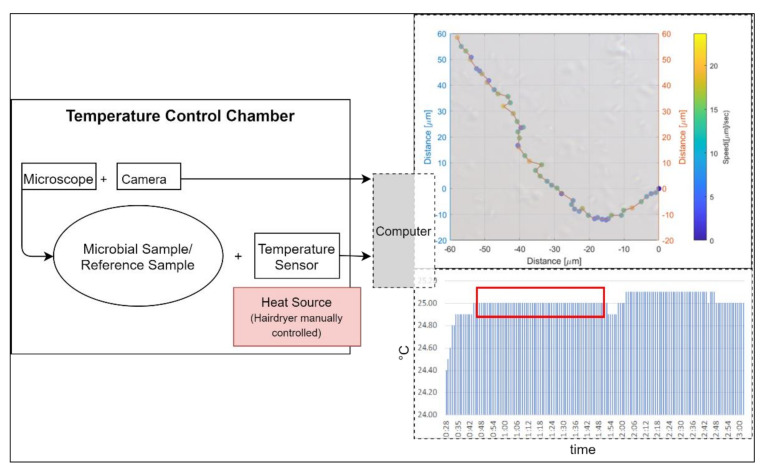
Setup of temperature control chamber: left side shows a schematic view, while the upper right image shows an example pathway of an *Escherichia coli* bacterium during a trial run at 25 °C. The lower right indicates the stability of the temperature for the same run over a time duration of 64 s (red box).

**Figure 2 life-11-00044-f002:**
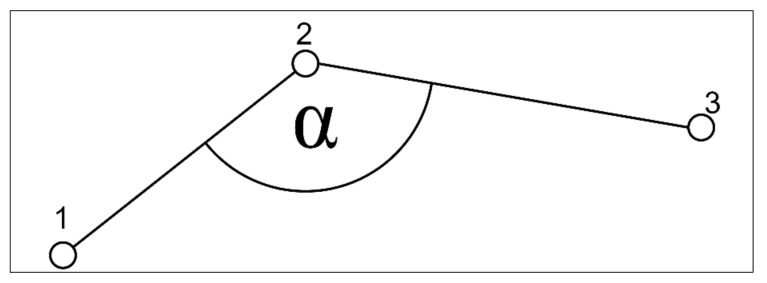
Sketch of angles of a microbe from three consecutive coordinates during motility behavior.

**Figure 3 life-11-00044-f003:**
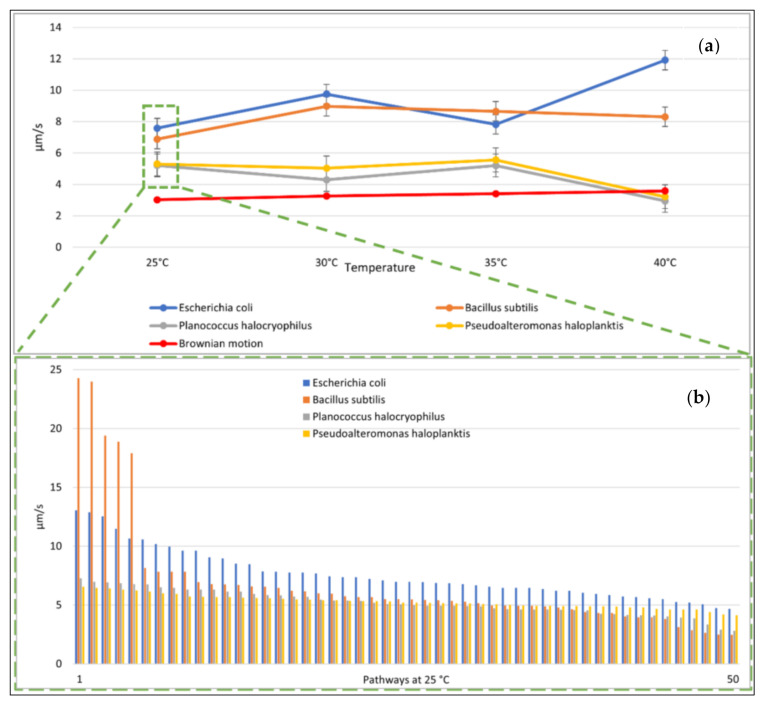
Mean microbial speed. (**a**) (top): Variations in mean microbial speed and particle mean speed due to Brownian motion as function of temperature with tracking accuracy. Note: The movement of the Brownian motion was simulated. (**b**) (bottom): Mean speed of each microbial pathway at 25 °C (sorted from fastest to slowest).

**Figure 4 life-11-00044-f004:**
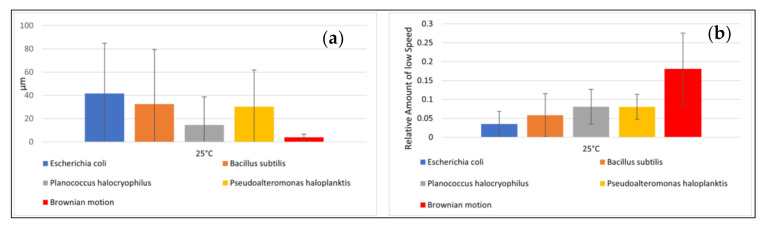
Motility data. (**a**) (left): Mean distance of microbial pathways and Brownian motion after ten seconds with standard deviation. (**b**) (right): Relative amount of low speed with standard deviation. Note: Brownian motion was simulated.

**Figure 5 life-11-00044-f005:**
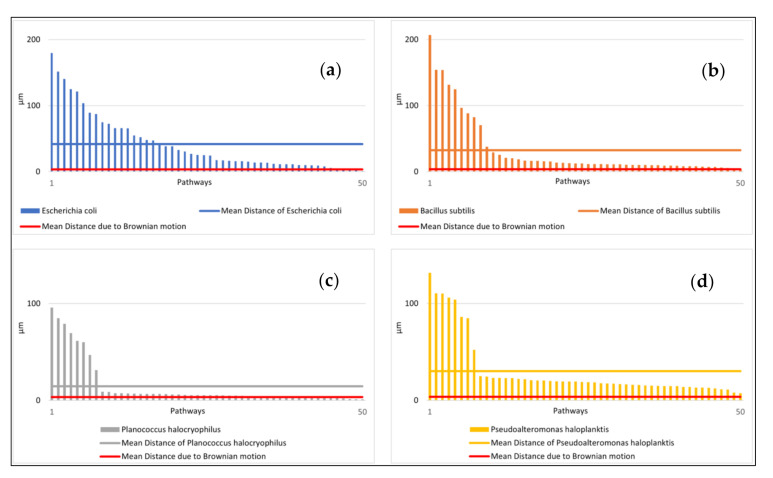
Distances of microbial pathways from their respective starting point after ten seconds at 25 °C (sorted from largest to smallest) in comparison to the mean distance due to Brownian motion. (**a**) (top left): Distances of the pathways of *E. coli*. (**b**) (top right), *B. subtilis*. (**c**) (bottom left): *P. halocryophilus*. (**d**) (bottom right): *P. haloplanktis*. Note: The movement of the Brownian motion was simulated. Its mean distance after ten seconds is shown in red.

**Figure 6 life-11-00044-f006:**
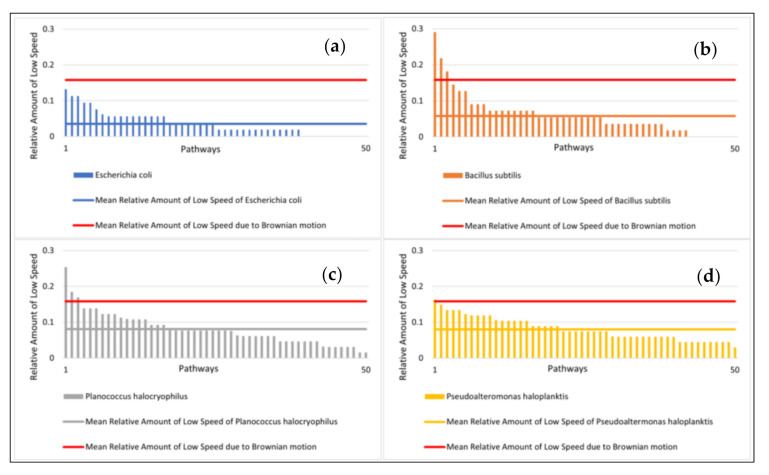
Relative amount of low speed at 25 °C with mean values (sorted from highest to lowest). (**a**) (top left): Relative amount of low speed of *E. coli*; (**b**) (top right): *B. subtilis*; (**c**) (bottom left): *P. halocryophilus*; and (**d**) (bottom right): *P. haloplanktis*. Note: The movement of Brownian motion was simulated. Its mean relative amount of low speed is shown in red. The feature low speed was here defined as speeds < 1.4 µm/s.

**Figure 7 life-11-00044-f007:**
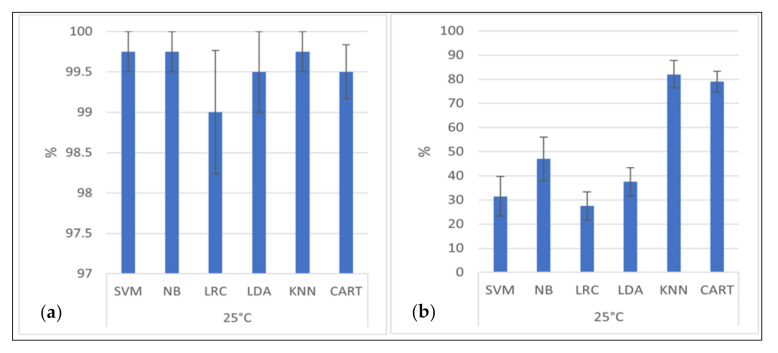
Classification results. (**a**) (left): Accuracy of the classification of biotic and abiotic particles with standard error. (**b**) (right): Accuracy of species identification with standard error. SVM is Support Vector Machines Classifier, NB is Naïve Bayes Classifier, LRC is Logistic Regression Classifier, LDA is Linear Discriminant Analysis Classifier, KNN is K-Nearest Neighbor Classifier, CART is Classification and Regression Tree Classifier.

**Table 1 life-11-00044-t001:** Algorithm performance vs. features used. Best performing feature selection for each algorithm shown in dark grey background. The KNN and CART Classifiers were found to be superior to the other algorithms, even with a non-optimal feature selection. SVM is Support Vector Machines Classifier, NB is Naïve Bayes Classifier, LRC is Logistic Regression Classifier, LDA is Linear Discriminant Analysis Classifier, KNN is K-Nearest Neighbor Classifier, CART is Classification and Regression Tree Classifier.

Classifier	Mean Speed of the Particle	Standard Deviation of Mean Particle Speed	Relative Amount of Clockwise Directional Change	Relative Amount of Counterclockwise Directional Change	Relative Amount of Low Directional Change	Average Directional Angle	Standard Deviation of Directional Changing Angles	Relative Amount of Low Speed	Mean Distance of Particles after Ten Seconds	Accuracy
KNN								**✓**		82%±5.7%
CART								**✓**		79%±4.3%
CART				**✓**						72%±3.7%
CART						**✓**		**✓**		71%±4.1%
CART				**✓**				**✓**		70%±4.5%
KNN				**✓**						68.5%±3.4%
KNN					**✓**					64.5%±4.4%
KNN			**✓**							63%±4.5%
NB	**✓**	**✓**	**✓**			**✓**	**✓**	**✓**		46.9%±9.1%
LDA	**✓**	**✓**	**✓**	**✓**	**✓**	**✓**		**✓**	**✓**	37.5%±5.8%
SVM	**✓**	**✓**								31.5%±8.2%
LRC	**✓**	**✓**	**✓**	**✓**		**✓**	**✓**		**✓**	27.5%±5.8%

## Data Availability

All data, classification algorithms and results, and the result and codes of the simulations that support the findings of this study are available online at http://dx.doi.org/10.14279/depositonce-10638.3.

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
