# Peer review of "Machine Learning Algorithms Applied to Identify Microbial Species by Their Motility"

_life, 2021, doi:10.3390/life11010044_

Round 1
Reviewer 1 Report
This is a useful and timely paper on the use of motility to identify living bacteria, using short microscopic recordings. I have only minor comments.
In Figure 3, both panels on the left hand side: The error bars are large and means are not very meaningful if the distributions aren't Gaussian. What do the distributions actually look like? Are they Gaussian or more bimodal? Please show/discuss.
The lower right panel should show error bars.
In the discussion, please mention the data volume required for each recording and compare it to expected data volumes in space missions.
In either the discussion or introduction, please mention the different types of bacterial swimming (e.g. run/tumble vs. run-reverse) and mention which category the selected strains belong to.
The language does require an editor as there are numerous instances of incorrect word use, tenses, etc.
Author Response
Dear Reviewer 1,
thank you very much for your helpful review!
Please find attached our replies to your comments.
Thank you very much again!
Sincerely,
Max Riekeles

Reviewer 2 Report
This is a longish list of notes to the authors - it's a mix of scientific comments, editorial recommendations, and editing notes, roughly in order of appearance in the text. I've turned it into a numbered list so it's easier for you to check them off.
Overall it's a valuable piece of research in an area that's very underdeveloped - the use of machine learning to evaluate scientific data with the eventual intent of using it to select data for transmission from space missions with limited data budgets. This is a nice application of ML to a current problem in the astrobiology community - deciding whether something seen in a microscopic image series is alive or not.
- Intro paragraph probably needs another sentence at the end that says “Here we address identification of motility with tools that can be extended to use in such missions” or something to that effect.
- End of first paragraph replace "criterium" with "criterion"
- Last paragraph, section 1 “optimal solution” should also address false positive/false negative rates (e.g, for a data prioritizer system you might prefer a bias toward false positives and let the ground operators decide)
- 11 µm/pixel sounds like a simple calculation based on nominal magnification but seems unreasonably high for an optical system – is this what you get from lambda/2NA? It looks like the stock 40x for the system you’re using is 0.65 NA, so if I use 550 nm as the middle of the visible range it should be about 0.325 micron resolution, or a little over a half wave, which makes more sense. So your resolution is lambda/2NA, but you have a pixel scale of 0.11 µm/px Note that later, in 2.4, when you’re estimating positions, it is fair to use the simple magnification based accuracy of 0.11 µm/px, which is really a scale, not resolution, and you can reasonably centroid objects to better than the optical resolution.
- If journal space allows, it would be helpful to include a diagram showing the temperature control chamber setup and an example time series of temperature data points indicating temperature stability. This could also be provided in the supplementary materials.
- Replace “insolating” with “insulating”
- Line 135: “recorded about 50 individual movement paths” - were there typically many more than 50 paths in each video? Approximately what fraction of the total number of paths was this? It’s reasonable to only do a fraction of the paths – hand tracking is extremely tedious and it’s not obvious that you get better training data beyond a certain point, but the authors should be clear about what fraction of paths in a dataset were tracked.
- Was frame rate limited entirely by shutter speed (as described) so theres no dead time between frames, or was the shutter time much shorter? If the former, you’d expect the bacteria to be smeared out and harder to localize in the images. It’s more typical for the camera shutter to be shorter than the inter-frame interval.
- Be consistent about how Fiji is capitalized (shows up as Fiji and FiJi in text). On the Fiji site they seem to stick with just uppercase “F”.
- In observing the reference solution to verify stability, was the sample of interest in the stage at the same time and just shifting back and forth so you could reasonably infer that it was stable?
- Line 180 change “simply” to “simple”
- Line 186-7: do the relative amount of clockwise and counterclockwise change need to be separate parameters? Could this just be done with a sign flip and use the same parameter for both?
- Line 205 it’s probably worth noting how this speed relates to the Brownian motion speed for a particles of interest (or note that it’s coming up in the text).
- At the start of 3.1, are the Brownian motion parameters from simulations and the rest from measured data on live bacteria? The use of “observed” suggests that it was experimental data, but it’s implied in the previous section that it’s probably simulated. That should be made more clear (probably split into two sentences, e.g. “The mean speed of Brownian motion in simulations was…” and “The observed mean speeds of the microbial species were…” or title the section “Motility Parameters from Hand Tracking” It would also be helpful to include the Brownian motion stats as a function of particle size in the supplementary material. What’s important is that this is slower than the motility you’re observing, but it would be useful to see it as a function of size for comparing with different bacteria & archaea. There are probably also too many significant figures shown in the speeds - the authors should review this.
- The authors should provide a few sentences of explanation as to why they didn’t use a non-motile species or killed bacteria as a Brownian motion reference – it would provide data that were taken under nearly identical conditions as a comparison for the motile species and validate the simulation. I’d recommend doing so if it wouldn’t be burdensome on the authors (lab access in many places is very limited because of pandemic restrictions), but expect that the results and conclusions won’t change as a result, though it would make it a stronger paper. The main value would be in showing the presence of any systematic effects on the motion, but the equilibration process the authors are using should be sufficient for the results here.
- Transition from 3.1 to 3.2 (check numbering – what should be “3.2 Results of Classification Algorithms” is numbered as “4.2 Results of Classification Algorithms”). It appears that 3.1 is the summary of the hand-tracked data, but a couple words at the start would make that unambiguous “The observed mean speeds of the hand-tracked microbial species were…”, then you jump right to the classification results without a transition to make clear that
- Figure 3 in 4.1 upper panels appear to be swapped – top left vertical axis is micron, but is supposed to be mean speed, while top right is supposed to be a distance but is units of micron/s
- Also the error bars in the mean distance traveled in 10s panel appear to be huge- it might be better to plot all the datapoints and show the means. Similarly with the temperature dependence of the speed - no error bars are shown, so either plot all the points or show the error bars.
- Referring to this set of four charts in early in the conclusions as an indicator of the value of using ML for this kind of data processing might be helpful- the data are very noisy and you are trying to develop a way to discriminate life from non-life with multiple independent criteria that are both hard to apply weights to and not easily traceable to basic physical principles.
- Section 4.2 (starting line 282): fix figure refs – they refer to fig 2a-b, should be fig 4 left/right
- Line 315 table 1: The dark grey background vs non-dark grey isn’t quite clear. It looks like the table should be titled “Algorithm performance vs. features used” or something like that, with the dark grey indicated as “best performing feature selection for each algorithm”
- Line 344, second paragraph of conclusions: restricting bacteria to 2D might also have made discrimination more difficult. g. if a turn was toward one of the surfaces it might have increased the apparent amount of “time at low speed” of a bacterium and made it come closer to falling into the Brownian class, even if it would have been still at higher speed in a 3D view. It will be interesting to see similar 3D experiments in the future.
- Lines 357-378 – Paragraphs should probably be reworded/rephrased to emphasize need to know and account for any external motions. E.g. stable laminar background flow can be subtracted off or otherwise accounted for in data processing, as can external stimuli like magnetic fields or light sources. These external disturbances can even be deliberately controlled to observe response of candidate organisms.
- Line 379 paragraph might work better in the intro section rather than conclusions, either right before or right after the second paragraph of the intro, with some rewording for flow
Author Response
Dear Reviewer 2,
thank you very much for your helpful review!
Please find attached our replies to your comments.
Thank you very much again!
Sincerely,
Max Riekeles
